# Elucidating the Onset of Cross-Protective Immunity after Intranasal Vaccination with the Attenuated African Swine Fever Vaccine Candidate BA71ΔCD2

**DOI:** 10.3390/vaccines12050517

**Published:** 2024-05-09

**Authors:** David Marín-Moraleda, Jordana Muñoz-Basagoiti, Aida Tort-Miró, María Jesús Navas, Marta Muñoz, Enric Vidal, Àlex Cobos, Beatriz Martín-Mur, Sochanwattey Meas, Veronika Motuzova, Chia-Yu Chang, Marta Gut, Francesc Accensi, Sonia Pina-Pedrero, José Ignacio Núñez, Anna Esteve-Codina, Boris Gavrilov, Fernando Rodriguez, Lihong Liu, Jordi Argilaguet

**Affiliations:** 1Unitat Mixta d’Investigació IRTA-UAB en Sanitat Animal, Centre de Recerca en Sanitat Animal (CReSA), Campus de la Universitat Autònoma de Barcelona (UAB), 08193 Bellaterra, Spain; david.marin@irta.cat (D.M.-M.); jordana.munoz@irta.cat (J.M.-B.); joseignacio.nunez@irta.cat (J.I.N.); 2Institut de Recerca i Tecnologia Agroalimentàries (IRTA), Programa de Sanitat Animal, Centre de Recerca en Sanitat Animal (CReSA), Campus de la Universitat Autònoma de Barcelona (UAB), 08193 Bellaterra, Spain; 3WOAH Collaborating Centre for the Research and Control of Emerging and Re-Emerging Swine Diseases in Europe (IRTA-CReSA), 08193 Bellaterra, Spain; 4Centro Nacional de Análisis Genómico (CNAG), Baldiri Reixac 4, 08028 Barcelona, Spain; 5Universitat de Barcelona (UB), 08034 Barcelona, Spain; 6School of Bioresources and Technology, King Mongkut’s University of Technology Thonburi, Bangkok 10150, Thailand; 7Department of Veterinary Medicine, National Chung Hsing University, Taichung 402, Taiwan; 8Biologics Development, Huvepharma, 3A Nikolay Haytov Street, 1113 Sofia, Bulgaria; 9Swedish Veterinary Agency (SVA), 751 89 Uppsala, Sweden

**Keywords:** ASFV, ASF immunity, live attenuated vaccine, LAV, BA71ΔCD2, onset of immunity

## Abstract

African swine fever (ASF) is a deadly disease of swine currently causing a worldwide pandemic, leading to severe economic consequences for the porcine industry. The control of disease spread is hampered by the limitation of available effective vaccines. Live attenuated vaccines (LAVs) are currently the most advanced vaccine prototypes, providing strong protection against ASF. However, the significant advances achieved using LAVs must be complemented with further studies to analyze vaccine-induced immunity. Here, we characterized the onset of cross-protective immunity triggered by the LAV candidate BA71ΔCD2. Intranasally vaccinated pigs were challenged with the virulent Georgia 2007/1 strain at days 3, 7 and 12 postvaccination. Only the animals vaccinated 12 days before the challenge had effectively controlled infection progression, showing low virus loads, minor clinical signs and a lack of the unbalanced inflammatory response characteristic of severe disease. Contrarily, the animals vaccinated 3 or 7 days before the challenge just showed a minor delay in disease progression. An analysis of the humoral response and whole blood transcriptome signatures demonstrated that the control of infection was associated with the presence of virus-specific IgG and a cytotoxic response before the challenge. These results contribute to our understanding of protective immunity induced by LAV-based vaccines, encouraging their use in emergency responses in ASF-affected areas.

## 1. Introduction

African swine fever (ASF), a severe hemorrhagic disease of compulsory notification to the World Organization for Animal Health (WOAH) (www.woah.org, accessed on 5 May 2024), is currently causing a pandemic, affecting both domestic and wild pigs. The disease is caused by the African swine fever virus (ASFV), a complex double-stranded DNA virus with a genome about 180 kbp in length from the *Asfarviridae* family. ASFV transmission is mediated by close contact between infected animals or through fomites and the ingestion of contaminated food, as well as by ticks from the *Ornithodoros* genus [1]. In terms of its most common clinical outcome outside Africa, ASF is characterized by a rapid and acute hemorrhagic disease, reaching up to 100% lethality and displaying a combination of signs such as intense fever, hemorrhages in various organs and pulmonary oedema [2]. The final stage of the disease is characterized by the massive apoptosis of lymphocytes [3] and an aberrant production of proinflammatory cytokines [4,5,6], reflecting an uncoordinated immune response caused by the massive virus expansion through the body. While ASF has remained endemic in Africa since its discovery, in 2007, the disease spread to many countries in Europe, Asia, Oceania and the Caribbean, provoking significant global economic losses in the swine industry [7]. Thus, the development of effective vaccines to control the current pandemic has become a major priority for the scientific community and pharmaceutical companies.

Live attenuated vaccines (LAVs) are, nowadays, the only available vaccine candidates inducing robust protective immune responses against ASF [8]. At present, there are numerous attenuated ASFV strains under research with one or more virulence genes deleted [9]. Although there is some concern regarding biosafety issues, these LAVs could become a valuable tool for controlling disease spread in ASF-affected areas. Indeed, the commercialization of the first ASF vaccine (based on the LAV ASFV-G-ΔI177L) [10] has recently been approved in Vietnam. Most of these LAVs have been generated from parental genotype II strains, conferring different degrees of homologous protection [11]. However, they fail to cross-protect against virulent genotype I strains that have recently also been identified in Asia [12]. To our knowledge, the genotype I-based BA71ΔCD2 LAV, developed by our group, is the only prototype providing robust cross-protective immunity against both ASFV genotypes I and II [13,14]. Indeed, this vaccine prototype confers solid protection against a lethal dose of the virulent ASFV Georgia 2007/1 (genotype II) 21 days after intranasal vaccination [14]. Regarding the immune responses induced by LAVs, in most cases, analyses have focused on virus-specific antibodies and IFNγ-producing cells without further characterization of the functional mechanisms correlating with protection. However, a recent study has associated the presence of neutralizing antibodies with the protection afforded by the LAV SFV-G-∆9GL/∆UK [15]. Also, we have shown that the protection induced by BA71ΔCD2 is associated with the induction of virus-specific Th1 and cytotoxic responses, as well as the prompt induction of innate immunity [14]. While all these studies contribute to the understanding of LAV-induced protective immunity, the onset of BA71ΔCD2-induced cross-protection has not been determined yet.

In the present study, we aimed to elucidate the onset of cross-protective immunity triggered by the ASF vaccine candidate BA71ΔCD2. The results demonstrate that vaccinated animals are able to control infection progression as early as 12 days postvaccination. This protection was concomitant with the presence of ASFV-specific IgG antibodies in sera, together with a blood transcriptomic signature associated with cellular cytotoxicity. Conversely, pigs challenged at days 3 or 7 postvaccination were not able to avoid systemic virus expansion and the appearance of clinical signs, only showing a delay in disease progression in some cases. These results indicate that early innate immunity induced by ASF LAVs is insufficient to confer solid protection against a lethal challenge, which highly depends on the onset of robust adaptive immune responses.

## 2. Materials and Methods

### 2.1. Ethics Statement

The animal experiment was performed in the biosafety level 3 facilities at the Centre de Recerca en Sanitat Animal (IRTA-CReSA, Barcelona, Spain). Animal care and procedures were conducted following the guidelines of Good Experimental Practice and were approved by the Ethics Committee on Animal Experimentation of the Generalitat de Catalunya (reference: CEA-OH/12072/1).

### 2.2. Viruses

BA71ΔCD2 is an LAV prototype derived from the virulent BA71 strain of African swine fever virus (ASFV), obtained by removing the CD2v gene (EP402R) through homologous recombination [13]. The highly virulent Georgia 2007/1 virus (genotype II) was generously provided by Dr. Linda Dixon from the World Organization for Animal Health (WOAH) reference laboratory at The Pirbright Institute, UK. The attenuated BA71ΔCD2 strain was expanded in the established COS-1 cell line (ATCC), while the virulent Georgia 2007/1 strain was produced in porcine alveolar macrophages obtained from lung lavage from healthy pigs (tested negative by PCR for porcine circovirus type 2, porcine reproductive and respiratory syndrome virus, and *Mycoplasma* spp.). BA71ΔCD2 and Georgia 2007/1 viruses were titrated with an immunoperoxidase monolayer (IPMA) [13] or hemadsorption assays [16], respectively. Titers were expressed as 50% tissue culture infectious dose (TCID50)/mL or 50% hemagglutination activity units (HAU50)/mL, respectively, according to the Reed and Müench method [17]. Then, a Poisson distribution was applied to convert the TCID50 titers to plaque-forming units (pfu) (TCID50/mL = 0.7 pfu/mL).

### 2.3. Animal Experiment

The animal experiment was performed with 20 seven-week-old Landrace x Large white breed male pigs. Before vaccination, animals were acclimated to the facilities over a minimum of seven days. Pigs were fed ad libitum throughout the length of the experiment. Pigs were separated into four groups of five pigs. Three groups were intranasally vaccinated with 10^6^ pfu/animal of the attenuated strain BA71ΔCD2 at 3, 7 or 12 days pre-challenge. Each animal received 2 mL of the vaccine diluted in PBS via intranasal inoculation (1 mL/nostril). As a control group, the remaining 5 animals were inoculated with 2 mL of PBS. Each animal was intranasally inoculated with 2 mL containing 10^5^ HAU50 of the highly virulent ASFV strain Georgia 2007/1 (1 mL/nostril). Clinical signs were monitored daily throughout the length of the experiment. EDTA blood, sera and nasal swabs were taken on the days of vaccination and the challenge, as well as at 3, 7 and 9 days postchallenge (dpc) to assess viral loads and host immune responses. Clinical signs were assessed according to previously defined standardized guidelines [18]. These included a daily evaluation of animal behavior, body condition (prominence of vertebrae and ribs), the presence of cyanosis, and digestive and respiratory signs. Post mortem examinations were conducted either to confirm or discard the presence of ASF-compatible pathological lesions. Each parameter was assigned a score from 0 to 4 based on severity (0: normal; 1: mild; 2: moderate; 3 and 4: severe).

### 2.4. Quantitative PCR for the Detection of ASFV

ASFV titers in blood and nasal swabs were assessed by a single-plex real-time quantitative polymerase chain reaction (qPCR). An IndiMag^®^ Pathogen Kit (INDICAL Bioscience, Leipzig, Germany) was used for the viral DNA extraction in blood and nasal swabs in a semi-automated manner by using a KingFisher System (Thermo Fisher Scientific, Waltham, MA, USA) according to the manufacturer’s instructions. To obtain the attenuated BA71ΔCD2 strain, the LacI gene was introduced by recombination in the parental BA71 ASFV strain to replace the CD2v gene [13]. Thus, BA71ΔCD2 and Georgia 2007/1 ASFV strains were identified targeting the LacI or CD2v genes, respectively. The primer pairs used for the detection of CD2v were 5′-CATGTTGAAGAAATAGAAAGTC-3′ and 5′-GCATGTAGTAAATAGGTGTATTAT-3′, for the amplification of an expected amplicon of 157 bp and its detection by the specific probe 5′-FAM-TAGGAAGTAATGGTTCTCTGGG-MGBNFQ-3′. LacI was amplified with the primer pair 5′-TCGGTACCCTCGACGGATTT-3′ and 5′-CGCGGGAAACGGTCTGATAA-3′ for an expected amplicon of 182 bp and its detection with the probe 5′-VIC-CTAGATGAAACCAGTAACGTTATAC-MGBNFQ-3′.

### 2.5. Enzyme-Linked Immunosorbent Assay (ELISA)

ASFV-specific IgG immunoglobulins were detected in pig sera by the WOAH-approved ELISA based on semi-purified cytoplasmic soluble antigen obtained as lysates of ASFV-infected COS-1 cells [19]. This same protocol was adapted to analyze ASFV-specific IgM immunoglobulins, as well as IgG1 and IgG2 subclasses. Briefly, antigen was diluted 1/1600 for plate coating. Sera samples were diluted (1/100 for IgM, 1/1000 for IgG and 1/100 for IgG1 and IgG2) in blocking buffer. Peroxidase-conjugated rabbit anti-pig IgG (1/20.000 dilution; Sigma-Aldrich, Saint Louis, MO, USA), goat anti-pig IgM (1/20.000 dilution; BioRad, Hercules, CA, USA), mouse anti-pig IgG1 (1/1000; BioRad), mouse anti-pig IgG2 (1/1000; BioRad) and peroxidase-conjugated goat anti-mouse IgG (A, G, M) (1/2500; Sigma-Aldrich) were used as secondary or detection antibodies for the presence of ASF-specific immunoglobulins. Reactions were performed using soluble 3,3′,5,5′-tetramethylbenzidine (TMB, Sigma-Aldrich) as a specific peroxidase substrate and stopped with 1N H_2_SO_4_. ELISA plates were read at 450 nm, and the results were represented as the average absorbance [optical density (OD) values] of duplicates.

### 2.6. Multiplex Luminex Assay

The Luminex xMAP technology was used to quantify the cytokine levels in sera, following the manufacturer’s instructions. The cytokine panel analyzed comprised IFNα, IFNγ, IL-1β, IL-4, IL-6, IL-8 (CXCL8), IL-10, IL-12/IL23p40 and TNFα (ProcartaPlex Porcine Cytokine and Chemokine Panel 1; Thermo Fisher Scientific). The ProcartaPlex Analysis App 1.0 software (Thermo Fisher Scientific) was used for data analysis and cytokine concentration determination (pg/mL), determined by each cytokine’s standard curve and median fluorescence intensities (MFIs).

### 2.7. RNA-Seq Library Preparation and Sequencing

Whole blood samples were stored at −80 °C in TRIzol Reagent (Invitrogen, Waltham, MA, USA) until required. Total RNA was obtained from whole blood using the phenol–chloroform method, followed by purification using the RNeasy Mini Kit (Qiagen, Hilden, Germany) following the manufacturer’s instructions. To ensure RNA quality, a DNase I treatment was performed using the RNase-Free DNase Set (Qiagen) for 15 min at room temperature. Total RNA extracted from whole blood was sent to the Centre Nacional d’Anàlisi Genòmica (CNAG; Barcelona, Spain) for sequencing. RNA was quantified with a Qubit RNA BR Assay kit (Thermo Fisher Scientific), and the RNA integrity was estimated using the RNA 6000 Nano Bioanalyzer 2100 Assay (Agilent, Santa Clara, CA, USA). The RNASeq libraries were prepared using the KAPA mRNA HyperPrep Kit (Roche, Basel, Switzerland), following the manufacturer’s recommendations, starting with 500 ng of total RNA as the input material. The quality of the library was controlled on an Agilent 2100 Bioanalyzer with the DNA 7500 assay. The libraries were sequenced on NovaSeq 6000 (Illumina, San Diego, CA, USA) with a 2 × 51 bp read length, following the manufacturer’s indications for dual indexing. Image analysis, base calling and quality scoring of the run were obtained by using the manufacturer’s software, Real-Time Analysis (RTA 3.4.4).

### 2.8. Bioinformatic Analysis

Illumina reads were mapped against a combined genome of Sus scrofa (Sscrofa11.1) and the African swine fever virus (ASFV) strain BA71V assemblies (accession number KP055815) using STAR software version 2.7.8a [20] with ENCODE parameters. Annotated genes were quantified with RSEM version 1.3.0 [21] with default parameters using a combination of the Sus scrofa ENSEMBL annotation release 110 and the KP055815 annotation. Subsequently, counts belonging to virus genes were filtered out. Differential expression analysis was performed with the limma v3.42.3 R package [22], using TMM normalization. The voom function [23] was used to transform the count data into log2 counts per million (logCPM), estimate the mean–variance relationship and compute observation-level weights. These voom-transformed counts were used to fit the linear models. Given the paired nature of the data, the individual variation was blocked using the duplicate correlation function. Contrasts for pairwise comparisons were extracted, as well as contrasts for the interaction effect between treatment and vaccination status. Genes were considered differentially expressed (DE) if they had an adjusted *p*-value < 0.05. Functional enrichment analysis was performed using DAVID (http://david.ncifcrf.gov/; accessed on 1 April 2024) [24], using all DE genes or the DE genes with an absolute fold change |FC| > 1.5, as specified in the corresponding figure legends. Sample similarities were inspected with a Principal Component Analysis (PCA) plot using the top 500 most variable genes. The virus presence in the samples was assessed with virus gene quantifications. Gene counts were normalized, log-transformed into logCPM and aggregated by sample. The aggregated logCPM values were presented using boxplots with the ggpubr R package, and group means were compared with a Wilcoxon test.

### 2.9. Statistical Analyses

The specific statistical analyses used are indicated in the figure footnotes of each dataset and were assessed by GraphPad Prism version 10.1.2. Statistical significance was set at *p* ≤ 0.05 (ns *p* > 0.05; * *p* ≤ 0.05; ** *p* ≤ 0.01; *** *p* ≤ 0.001). Graphs were created with GraphPad Prism version 10.1.2. software, R 4.3.3 and RStudio version 2022.07.2 software and the BioRender online tool.

## 3. Results

### 3.1. Onset of Cross-Protection Induced in Pigs Intranasally Vaccinated with BA71ΔCD2

We have previously demonstrated that a single intranasal dose of 10^6^ pfu of BA71ΔCD2 confers solid protection against a lethal infection with the ASFV strain Georgia 2007/01 at 21 days postvaccination [14]. Here, we aimed to elucidate the onset of this cross-protection by vaccinating animals (five pigs/group) 12 days (12 dv), 7 days (7 dv) and 3 days (3 dv) before an intranasal challenge with the virulent strain Georgia 2007/01 (10^5^ HAU). A group of five pigs was left unvaccinated as the control (Figure 1A). The animals were monitored daily to observe clinical signs and sacrificed at 9 days postchallenge (dpc) to further observe ASF-compatible macroscopic lesions. All the unvaccinated animals synchronously succumbed to the disease at 8 dpc (Figure 1B), demonstrating the validity of the lethal challenge model used. In contrast, on the day of sacrifice (9 dpc), two out of five pigs from the 3 dv group and all the animals from the 7 dv and 12 dv groups were still alive, indicating that the early immune response induced by BA71ΔCD2 is at least capable of delaying disease progression (Figure 1B). Daily measured temperatures and necropsy examinations correlated with the survival trends observed (Figure 1C,D). The pigs from the 3 dv group presented the same temperature pattern as the unvaccinated animals and similar severe lesions in most tissues, characteristic of acute ASF disease. Interestingly, the 7 dv group showed a slight delay in temperature rise and more moderate lesions, which were mainly concentrated on lymph nodes, suggesting that vaccine-induced early immunity reduces disease severity. In clear contrast, the animals from the 12 dv group showed rectal temperatures below 41 °C, and the numbers of animals and tissues affected by ASF lesions were much lower than in the other groups. These results indicate that animals vaccinated with BA71ΔCD2 12 days before a challenge can control a lethal ASFV infection (Figure 1C,D).

Next, viral DNA in blood and nasal swab samples was quantified to assess the ability of each vaccination schedule to reduce ASFV viremia and virus shedding (Figure 2). To independently detect BA71ΔCD2 and Georgia 2007/1 virus strains, the gene LacI (which replaces CD2v in the attenuated strain) or the gene CD2v was quantified by qPCR, respectively. BA71ΔCD2 genomic equivalent copies (GECs) were found in nasal swabs in some pigs from day 6 to day 15 postvaccination, indicating the local replication of the vaccine virus. On the contrary, the BA71ΔCD2 levels in blood were below the limit of detection, except for one single pig at 9 dpc (Figure 2). In the samples obtained postchallenge, Georgia 2007/1-related viremia was evident from day 3 dpc in the unvaccinated and 3 dv groups and from day 7 dpc in the 7 dv group (Figure 2) and correlated with the rise in temperatures (Figure 1C). The three groups also showed shedding at different time points postchallenge, with a slight delay in those animals vaccinated earlier (Figure 2). Importantly, group 12 dv was the only group clearly controlling Georgia 2007/1 replication (Figure 2). These animals showed a lack of virulent virus shedding throughout the experiment, and only one pig was positive in blood at 9 dpc. Altogether, these results demonstrate that intranasal vaccination with BA71ΔCD2 12 days before the lethal challenge is effective in abrogating virus replication and dissemination through the body and viral shedding after an intranasal challenge.

### 3.2. Presence of ASFV-Specific IgG Antibodies before Challenge Is Associated with Control of Infection

Despite the immunological mechanisms behind the protective role of ASFV-specific antibodies remaining elusive, the appearance of serum IgG antibodies after LAV-based immunizations often coincides with protection [14,25,26]. However, to the best of our knowledge, the correlation of virus-specific IgM antibodies with protection has never been evaluated. Thus, ASFV-specific IgM and IgG antibodies were quantified by an ELISA throughout the length of the experiment (Figure 3). On the day of the Georgia 2007/1 challenge, all the animals vaccinated 3 days before (3 dv) were negative for IgM antibodies, like the unvaccinated control group (Figure 3A). Contrarily, three out of five pigs from group 7 dv and all the animals from group 12 dv showed high levels of ASFV-specific IgM antibodies (Figure 3A). However, the presence of IgM antibodies did not correlate with the control of infection since the IgM-negative pig #10 (group 7 dv) had minor ASF lesions as well as low fever and virus loads (Figure 1 and Figure 2). Interestingly, at 3 dpc, all the pigs from group 7 dv became positive for ASFV-specific IgM, indicating a booster effect induced by the Georgia 2007/1 infection. In contrast, IgM antibodies decreased after the challenge in the pigs from group 12 dv, suggesting a lack of productive Georgia 2007/1 infection and a corresponding antibody boost in these animals. Regarding virus-specific IgG antibodies, only animals from group 12 dv were positive on the day of the challenge, while a gradual increase was observed postchallenge in pigs from group 7 dv (Figure 3B). The analysis of the IgG subclasses induced in group 12 dv at 0 dpc showed a balanced IgG1/IgG2 ratio (Figure 3C). Finally, the pigs from group 3 dv showed a very slight increase in virus-specific IgG antibodies at 7 dpc, with significantly higher titers when compared to the unvaccinated group (Figure 3D), suggesting a link with the minor delay in disease onset in this group. These results further indicate an association between the presence of IgG antibodies and the retardation of ASF clinical signs in vaccinated animals.

### 3.3. Lack of ASF-Associated Apoptosis and Immunopathology Markers in Pigs Vaccinated 12 Days Prior to the Challenge

To gain better insight into the immunological context associated with the different infection outcomes, whole blood transcriptomic signatures from days 0, 3 and 7 postchallenge by RNA sequencing (RNA-seq) were obtained. The Principal Component Analysis (PCA) of the datasets obtained revealed major transcriptional changes in the samples from the unvaccinated and 3 dv groups at 7 dpc, in which acute ASF disease was evident. Moreover, validating the correlation of these transcriptional changes with the animal health status, the three animals from group 7 dpc showing severe ASF lesions in tissues (pigs #7, #8 and #9; Figure 1D) were also present in this cluster (Figure 4A). Taking samples from day 0 postchallenge as a reference, differentially expressed (DE) genes with an absolute fold change |FC| > 1.5 were obtained in the four groups at each time point analyzed. In line with the results from the PCA, both the 3 dv and unvaccinated groups showed a higher number of DE genes at 7 dpc (757 and 1342, respectively), corresponding to the animals showing more severe ASF clinical signs (Figure 4B). In contrast, the number of DE genes in groups 7 dv and 12 dv was much lower at all time points. These results indicate that the whole blood transcriptome better reflects pathological events than protective immune responses. Indeed, the Gene Ontology (GO) analysis showed enriched terms related to inflammation and innate immunity in group 3 dv and in the unvaccinated group at both 3 dpc and 7 dpc (Figure 4C). In contrast, group 7 dv did not show enriched terms at any time point, probably due to the low number of DE genes with an absolute fold change |FC| > 1.5. Interestingly, in group 12 dv at 3 dpc, enriched terms related to innate immunity as well as IL-2 cytokine production were also identified (Figure 4C). These results are in line with our previously found association of the prompt activation of Th1 cells and a concomitant inflammatory response with BA71ΔCD2-induced protection [14].

To corroborate that the animal health status of each experimental group was reflected in their respective whole blood transcriptomes, we next focused on the analysis to identify a pathology-associated transcriptomic signature. Thus, DE genes between 7 dpc and 0 dpc from the unvaccinated pigs were further analyzed, and their expression levels were compared among the different groups. From this analysis, genes representative of relevant immune processes and pathways are shown in Figure 5. In concordance with other studies [5,27], at 7 dpc, the unvaccinated pigs showed higher expression levels of genes related to inflammation and innate immunity. However, this marked difference was not evident at 3 dpc. Importantly, several type I interferon (IFN)-stimulated genes (ISGs) were strongly associated with disease progression, showing at 7 dpc a gradual upregulation depending on the time of vaccination before the challenge. A similar expression pattern was observed for several inflammatory mediators, cytokines, chemokines and cell receptor and activation markers (Figure 5). Of note, other genes showed an opposite pattern, being downregulated in the unvaccinated pigs compared to group 12 dv, indicating its potential role in protection. This might be the case for IRF5, a marker of inflammatory macrophages in vivo [28], which also plays a critical role in the development of a Th1 response [29]. Several genes of the MAPK pathway were also downregulated in the unvaccinated animals, suggesting that this pathway is not involved in the inflammatory response associated with acute ASF disease. Other genes with a similar expression pattern illustrate the characteristic severe leukopenia of acute ASF infection. For instance, in the diseased groups, the B cell markers CD79A and CD79B, the T cell marker CD28, genes related to antigen presentation (SLAs, CD74), and the macrophage receptor CSF1R were also downregulated. Finally, genes related to apoptosis (MAPK14, CASP1-3-7-8), also characteristic of acute ASF infection, showed a clear association with a lack of protection. Indeed, expression levels at 7 dpc of several caspases were similarly upregulated in all the groups except for animals from group 12 dv. Overall, this expression pattern demonstrates that the time of vaccination before viral infection clearly influences the kinetics of the immune system response to virus exposure.

To further validate the lack of systemic inflammation in the pigs vaccinated 12 days before the challenge, the levels of relevant cytokines in sera between the unvaccinated pigs and the 12 dv group were compared. Figure 6 shows the results obtained by the multiplex Luminex assay from samples from the 0 dpc, 3 dpc and 7 dpc groups. Except for IL-6 and IL-8, several proinflammatory and anti-inflammatory cytokines showed a tendency to increase after the challenge in the unvaccinated pigs. However, only IFNa, IL-12 and IL-10 showed significant differences between the 7 dpc and 0 dpc groups. In line with the results obtained by RNA-seq, the pigs from group 12 dv showed background levels of all cytokines without evident differences before and after the challenge. Finally, an analysis of ASFV counts in blood obtained by RNA-seq further validated the lack of viremia in the pigs vaccinated 12 days before the challenge, as well as significantly lower levels of viral transcripts in group 7 dv (Figure 7). Altogether, these results demonstrate that vaccination 12 days before the challenge is enough to confer protection against a lethal challenge.

### 3.4. Control of Infection Is Associated with a Blood Cytotoxic Transcriptomic Signature before Challenge

The early protection observed in the vaccinated pigs from group 12 dv can be determined by ongoing immune mechanisms present at the time of challenge. Therefore, the RNA-seq dataset was next analyzed to identify the DE genes at 0 dpc in each vaccinated group compared to the unvaccinated controls. Interestingly, only the 12 dv group showed a significant number of DE genes (2862), while only four DE genes were detected in group 3 dv and none in group 7 dv (Figure 8A). The analysis of these genes in the 12 dv group revealed that, at the moment of the lethal challenge, these pigs had an ongoing vaccine-induced adaptive immune response. This immunity involved both innate and adaptive components, mainly related to cellular responses. Figure 8B shows several representative DE genes of these immune mechanisms present at the onset of protection. Some DE genes related to inflammation were still upregulated 12 days postvaccination. But, more importantly, a higher expression of genes related to cytotoxic cell responses clearly differentiated the protected pigs in the 12 dv group from the animals in the other groups suffering from the disease. The identification of genes related to both CD8 T and NK cells, together with cytotoxic markers such as perforin and granzymes, suggests that both immune cell subsets might have been present in the blood of the protected pigs at the time of challenge. Interestingly, all these genes were also upregulated in the unvaccinated pigs at day 7 postchallenge, further indicating the major role of a prompt cytotoxic response to control the course of ASFV infection [14].

## 4. Discussion

Live attenuated vaccines (LAVs) are the only current vaccine candidates that induce solid protective immunity with the capacity to control the present ASF pandemic. Indeed, very recently, the use of the first vaccine against ASF was approved in Vietnam [10]. While there are high expectations for the progress of this vaccination strategy in the field, there is hope that LAVs might become a useful tool as emergency vaccines to reduce disease spreading among farms from ASF-affected countries. However, our understanding of LAV-induced protective immunity in pigs remains very limited and requires further investigation. Here, we established the onset of protection induced by the intranasal inoculation of the vaccine candidate BA71ΔCD2. We demonstrated that solid protective immunity is established at 12 days postvaccination, as demonstrated by the absence of significant clinical signs, low virus loads and a lack of an unbalanced inflammatory response after the challenge with the virulent Georgia 2007/1 strain. In contrast, pigs vaccinated 3 or 7 days before the challenge were not fully protected, only showing a delay in disease progression. Animals controlling the lethal infection showed the presence of systemic adaptive immune responses, characterized by the presence of virus-specific IgG antibodies in serum as well as markers of cytotoxic cells in blood. These results contribute to our understanding of how ASF LAVs function and provide further evidence of immune mechanisms that might play a critical role in protection against ASF.

There are few other studies investigating the onset of protection induced by ASF LAVs [10,25,26]. However, the LAVs tested and the experimental approaches used have significant differences from the present study. First, BA71ΔCD2 is the only LAV capable of conferring heterologous protection against the two virulent genotypes I and II circulating in Asia [12,14,30]. Second, the use of an intranasal ASFV lethal infection better mimics the naturally occurring infections in the field, contrasting with the standardized intramuscular challenges used in the other studies. Thus, the results presented here extend those previous findings and contribute to a better understanding of LAV-induced protection. Nevertheless, our results are in line with the other LAVs used, showing in all cases a progressive acquisition of protective immunity after vaccination, which is associated with the appearance of adaptive immune responses. Importantly, we provide solid evidence of a high protection rate at 12 days postvaccination with BA71ΔCD2, in contrast with the ASFV-G-ΔI177L vaccine, which protected 50% of the pigs at a similar time point [10]. This difference might be explained by the vaccine doses used in each study, as well as the differing vaccine administration routes and challenge models. Unfortunately, due to space limitations in our BSL3 facility, we had to end the study at day 9 postchallenge, and thus we could not further validate the robustness of the protection achieved concerning long-term survival. Nevertheless, the low virus loads and minor clinical symptoms demonstrate that these animals did control virus expansion. This conclusion is further supported by our previous studies using BA71ΔCD2. Indeed, we have observed robust protection in pigs inoculated even with a lower vaccine dose, which had higher rectal temperatures in sera and similar virus titers after the challenge as those observed in the present study [13,14]. Further evidence of this solid protection was obtained through the transcriptomic analysis, showing a lack of apoptosis markers and an unbalanced inflammatory response at later time points, two characteristic features of acute ASF infection [4,5,6]. Altogether, these results ensure that animals vaccinated 12 days before a challenge would survive the lethal infection without clinical manifestation. Thus, this study contributes to the characterization of the onset of protective immunity induced by ASFV LAVs, a critical issue that needs solving to validate their use as a vaccine emergency response, as already demonstrated for food-and-mouth disease and classical swine fever disease vaccines [31,32,33,34,35].

The identification of reliable immune correlates of protection for ASF remains elusive. In the present study, the analysis of humoral response kinetics showed that IgM antibodies were not associated with protection, resulting in a mere indicator of early vaccination or infection. In contrast and in line with previous works using other LAVs [10,26], we showed the presence of a high level of ASFV-specific IgG antibodies at the time of the challenge in the protected animals. Moreover, the animals challenged 3 and 7 days postvaccination showed a significantly higher titer of IgG antibodies at day 7 postchallenge compared to the unvaccinated pigs. This minor boost might explain the small delay in disease progression observed in these vaccinated pigs and thus confirm the potential role of virus-specific antibodies in controlling virus expansion. However, it has been shown that an early antibody response measured by an ELISA is not a good correlate of protection [25]. More exhaustive analyses should be performed to identify antibody-mediated functional responses better correlating with protection. Indeed, the role of humoral responses in controlling ASF was demonstrated by the in vivo inoculation of immunoglobulins or colostrum from immune pigs [36,37], probably mediated by IgG antibodies. In line with this, a recent study has shown a significant association between neutralizing antibodies and protection in SFV-G-∆9GL/∆UK-vaccinated pigs [15]. However, this neutralization assay is based on the use of a virus strain highly adapted to Vero cells, which probably does not reflect the actual physiological conditions in the context of a virulent ASFV strain. Nevertheless, these results are very encouraging and provide evidence of a potential method to evaluate vaccine efficacy without the need to challenge vaccinated pigs. Further studies are required to investigate the presence of neutralizing antibodies and their correlation with protection against homologous and heterologous strains in BA71ΔCD2-vaccinated animals.

We also identified a transcriptomic signature related to cytotoxic responses in the blood of the protected pigs at the time of the challenge. Together with the cytotoxic markers such as perforin and granzymes, the results showed the upregulation of genes related to both NK and T cells. These results are in line with previous studies associating the presence of perforin-producing NK and T cells in blood with the control of ASFV systemic infection [38,39,40]. Also, we have recently demonstrated the expansion of NK cells, single CD8+ T cells, and double positive CD4 + CD8+ T cells during in vitro recall responses of blood cells from pigs three weeks after BA71ΔCD2 vaccination [14]. It is reasonable to hypothesize that the results from the present study reveal the presence of these cytotoxic cells during the peak of the adaptive immune response, which are only detectable one week after through an in vitro-specific stimulation. Of note, the unprotected pigs also showed an upregulation of this transcriptomic signature at 7 dpc (Figure 8B), when the acute disease was evident. This suggests that a systemic cytotoxic response might be associated with both protection and immunopathology, depending on the context. The identification of the different cytotoxic cell subsets involved in each of these opposed contexts will be necessary to clarify this issue. Finally, it is important to mention that cytotoxic T cells might have a relevant role in the cross-protection induced by BA71ΔCD2 vaccination, as already demonstrated in other models such as Influenza A infection [41,42]. Further studies are required to analyze the relative importance of cytotoxic T cells in fighting against both homologous and heterologous ASFV infections. Furthermore, a more ambitious study testing a large number of protected and unprotected animals is necessary to clarify whether any of these cytotoxic cell subsets could be a good correlate of protection against ASF. Nonetheless, the cumulative data from several studies indicate that this immune mechanism is crucial for conferring protection against ASF [43] and hence an essential immune target when developing novel vaccines.

## 5. Conclusions

In conclusion, this work helps to better understand the protective immunity induced by LAVs against ASF. We demonstrate that the intranasal administration of the BA71ΔCD2 LAV confers cross-protection against the pandemic Georgia 2007/1 strain 12 days after vaccination. Also, we provide evidence of potential humoral and cellular immune correlates of protection to be further evaluated. Importantly, due to the intranasal challenge model used, the results from this study represent a proof of concept for what might actually occur during vaccination programs on farms. These findings are relevant in the present pandemic context, where attenuated ASFVs are becoming a strategic tool as a vaccine emergency response in ASF-affected areas.

## Figures and Tables

**Figure 1 vaccines-12-00517-f001:**
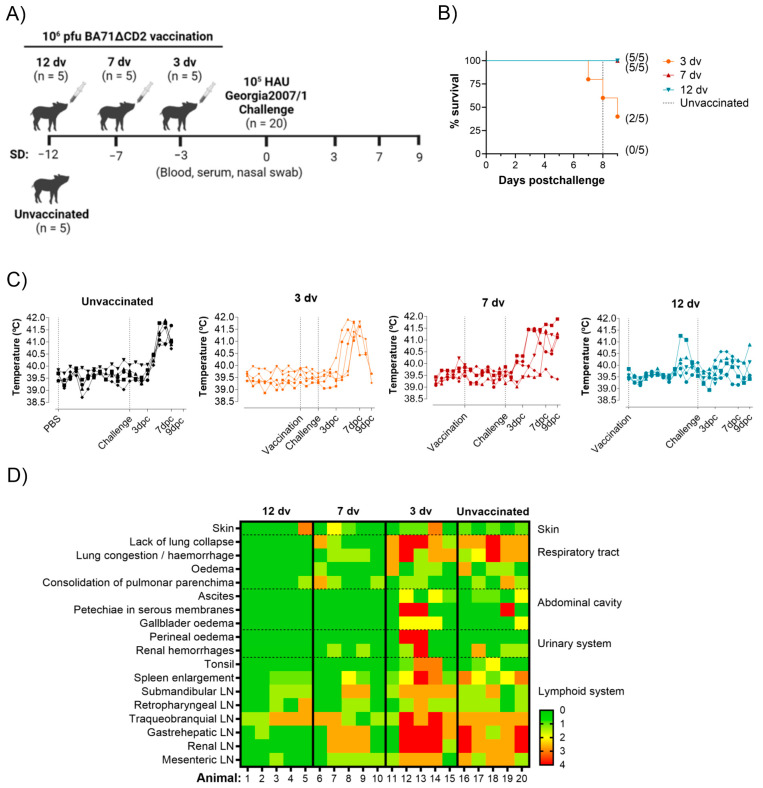
Determination of the onset of protection after intranasal vaccination with BA71ΔCD2. (**A**) Schematic representation of the experimental design. (**B**) Survival plot showing the percentage of alive animals at the indicated time points after intranasal challenge with Georgia 2007/1. The number of surviving pigs at the end of the experiment is indicated in brackets. (**C**) Rectal temperature charts from individual pigs throughout the experiment. Vertical dotted black lines indicate the days of the vaccination and challenge. (**D**) Macroscopic lesion scores of each animal at necropsies. The severity of the lesions is represented by a gradient between green (absence), yellow (mild), orange (moderate) and red (severe). For the “Petechiae in serous membranes” and “Perirenal oedema” categories, only the presence or absence of lesions is represented. LN: lymph node.

**Figure 2 vaccines-12-00517-f002:**
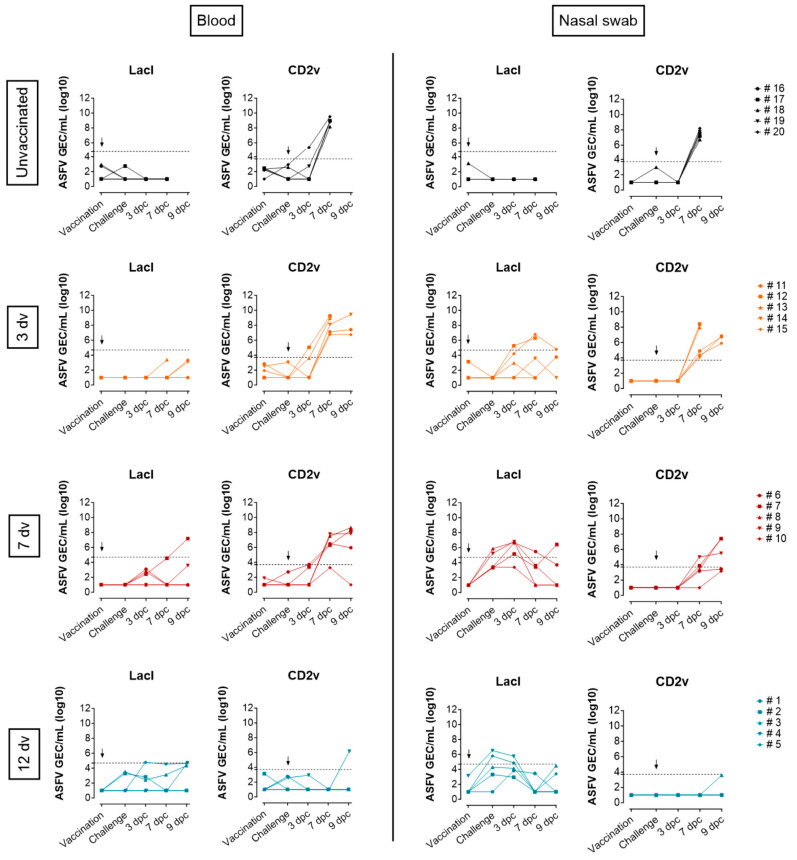
Pigs vaccinated 12 days before the ASFV lethal challenge control viremia and viral shedding. Virus titers in whole blood and nasal swabs measured by qPCR at the indicated time points after intranasal vaccination and challenge. Genomic equivalent copies (GECs) of ASFV were quantified by the detection of the gene LacI, which replaces CD2v in the attenuated BA71ΔCD2 strain, and the CD2v gene, which is encoded in the virulent Georgia 2007/1 genome. The arrows represent the vaccination (for LacI) or challenge (for CD2v) time points. The black dotted lines represent the limit of detection of each assay, and values below are considered indeterminate or negative. Symbols represent the mean of duplicates.

**Figure 3 vaccines-12-00517-f003:**
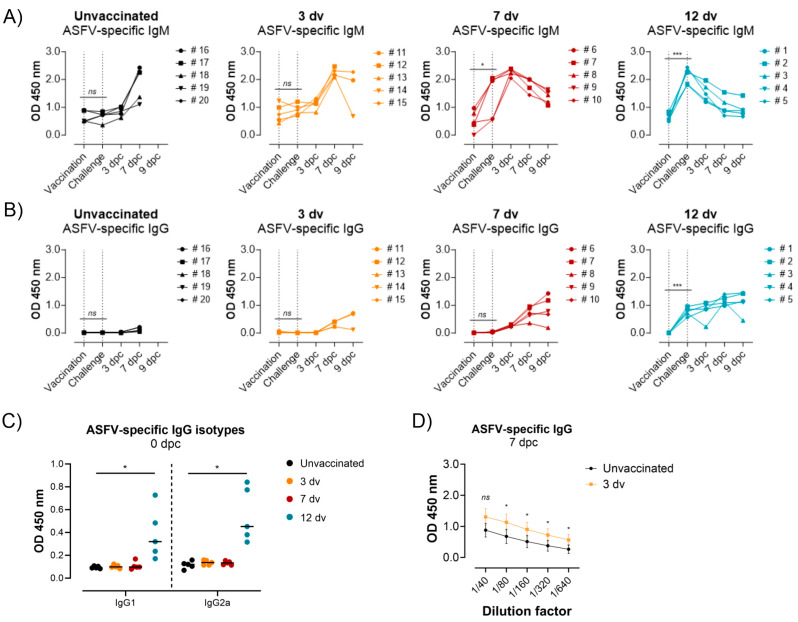
The presence of ASFV-specific IgG before the challenge, but not IgM, is associated with the control of infection. The induction of ASFV-specific IgM (**A**) and IgG (**B**) antibodies was measured for each animal by an ELISA at the indicated time points. Black dotted lines represent the vaccination and challenge time points. (**C**) Analysis of ASFV-specific IgG1 and IgG2 isotypes in sera (1/100 dilution) at 12 days postvaccination. (**D**) ASFV-specific IgG antibody titers were compared between unvaccinated animals and animals vaccinated 3 days prior to the challenge. The mean and standard deviation of the average IgG levels from each group and time point are represented. Statistical significance was assessed by a paired *t*-test (ns *p* > 0.05; * *p* ≤ 0.05; *** *p* ≤ 0.001).

**Figure 4 vaccines-12-00517-f004:**
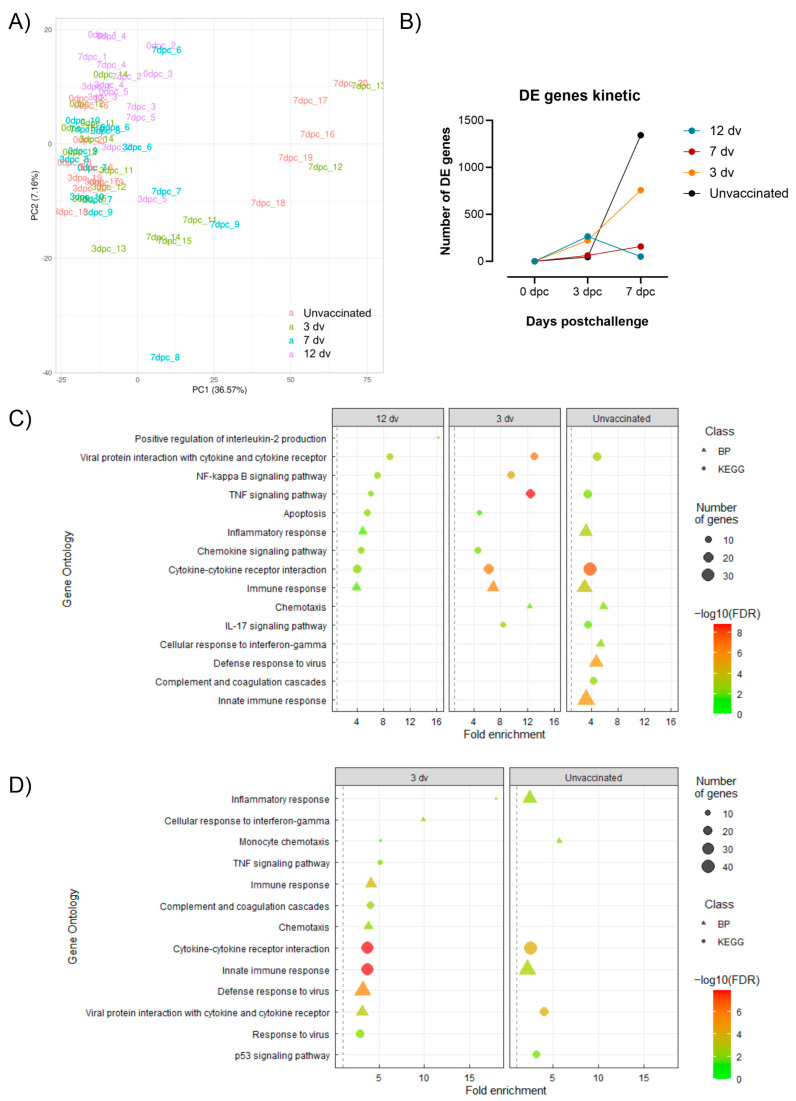
Time-resolved whole blood transcriptomic changes after Georgia 2007/1 infection. (**A**) PCA of the normalized RNA-seq expression levels (log_2_CPM) of the top 500 most variable genes. (**B**) Number of DE genes with an absolute fold change |FC| > 1.5 per time point in each group compared to 0 dpc. (**C**,**D**) List of representative Gene Ontology (GO) terms enriched in DE genes with an absolute fold change |FC| > 1.5 from vaccinated and unvaccinated pigs at 3 dpc (**C**) and 7 dpc (**D**). The size of the dots represents the number of DE genes associated with the GO term; the shape indicates the GO database (BP; biological processes) or the Kyoto Encyclopedia of Genes and Genomes (KEGG); and the color indicates the negative log10 value of the false discovery rate (FDR).

**Figure 5 vaccines-12-00517-f005:**
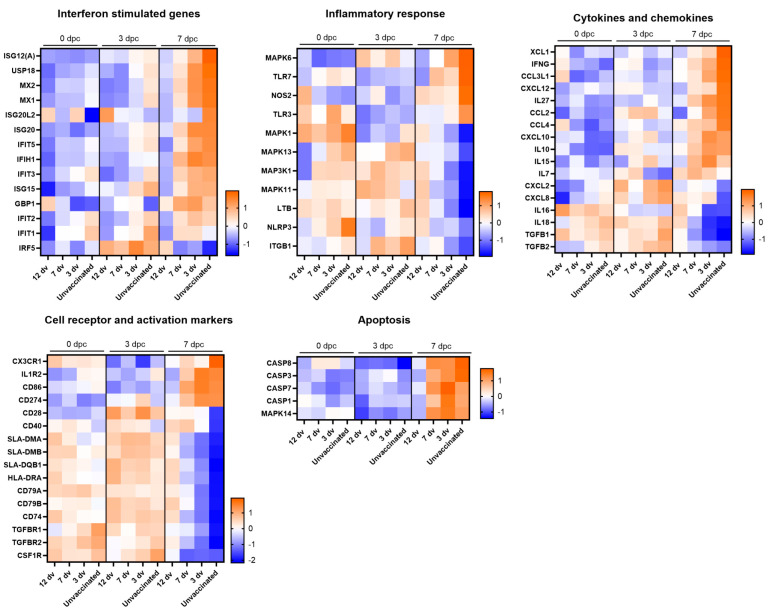
Whole blood transcriptomic signatures reveal the lack of ASF-associated apoptosis and immunopathology markers in pigs vaccinated 12 days before the challenge. Heatmaps depict normalized RNA-seq-derived log_2_CPM values of representative DE genes in the categories indicated.

**Figure 6 vaccines-12-00517-f006:**
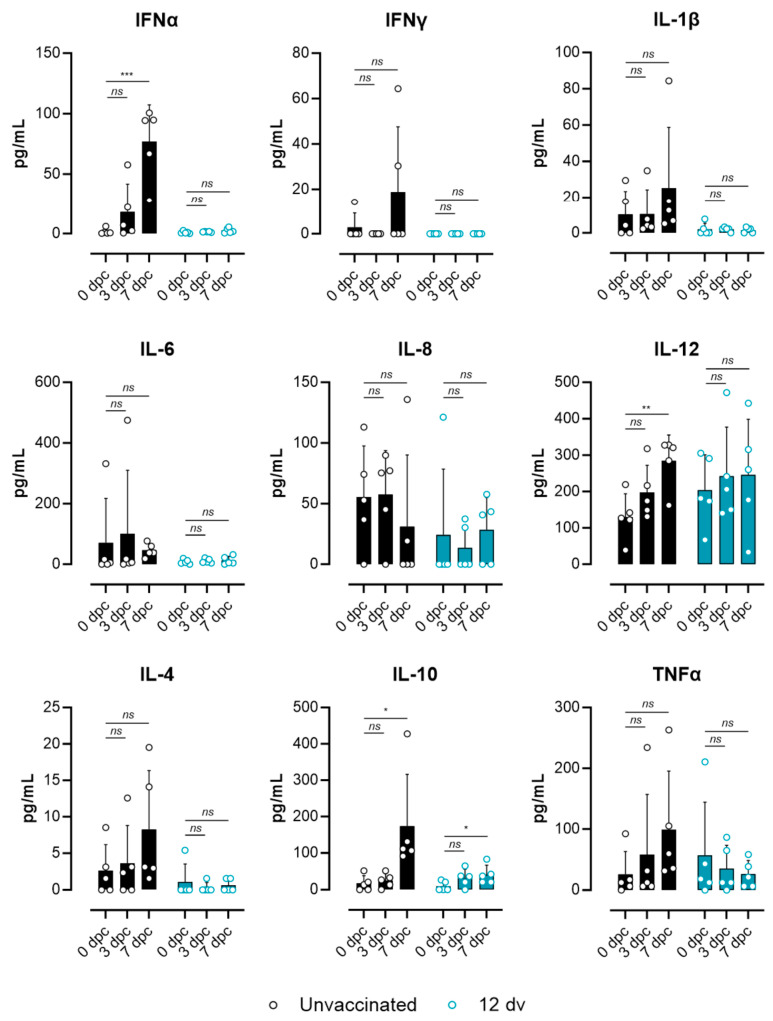
Lack of an inflammatory-associated cytokine signature at day 9 postchallenge in pigs vaccinated 12 days before the challenge. Cytokine levels in sera from unvaccinated (black symbols) and 12 dv group animals (blue symbols) at the indicated time points were quantified by a Luminex-based multiplex assay. The mean and standard deviations from each group are represented. Statistical significance was determined by an unpaired *t*-test (*p* > 0.05 ns; * *p* ≤ 0.05; ** *p* ≤ 0.01; *** *p* ≤ 0.001).

**Figure 7 vaccines-12-00517-f007:**
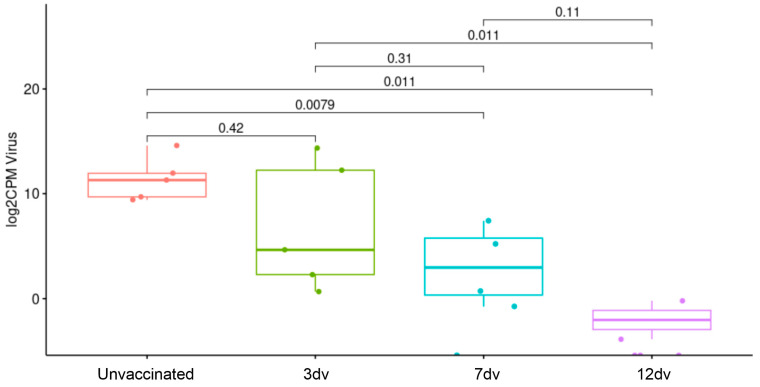
RNA-seq-derived number of counts mapping to ASFV genes in blood from unvaccinated and vaccinated pigs. Statistical significance was determined by a Wilcoxon test (*p*-values are shown).

**Figure 8 vaccines-12-00517-f008:**
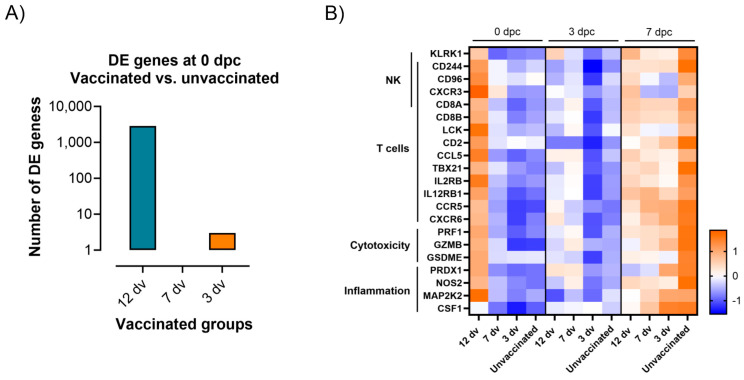
Differentially expressed genes associated with protection against ASFV in vaccinated and unvaccinated pigs before the challenge. (**A**) The number of differentially expressed (DE) genes in each vaccinated group compared to the unvaccinated group at 0 dpc. (**B**) Heatmap depicting normalized RNA-seq-derived log_2_CPM values of representative DE genes in the categories indicated.

## Data Availability

All raw and processed sequencing data generated in this study have been submitted to the NCBI Gene Expression Omnibus database (GEO; http://www.ncbi.nlm.nih.gov/geo/, accessed on 5 May 2024) under accession number GSE266565.

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
