# Peer review of "Elucidating the Onset of Cross-Protective Immunity after Intranasal Vaccination with the Attenuated African Swine Fever Vaccine Candidate BA71ΔCD2"

_vaccines, 2024, doi:10.3390/vaccines12050517_

Round 1

Reviewer 1 Report

Comments and Suggestions for Authors

In fact, this paper is a sequel to a previous paper (Journal of Virology, 2017, 91(21):e01058-17). Unfortunately, the results of this paper are not as ideal as described in the previous paper. Especially the Genomic equivalent copies (GEC) testing after challenge (Figure 2). As ASFV vaccine, even if only one pig shows viral shedding (wild virus) after challenge, the entire immunity still fails, as this can lead to an ASF epidemic in the swine farm. So, I don't think this is an ideal vaccine.

Minor, line 115, "106 pfu/animal", line 119, "105 HAU50", 6 and 5 should be superscript.

Reviewer 2 Report

Comments and Suggestions for Authors

This study characterized the onset of cross-protective immunity triggered by the LAV candidate BA71ΔCD2, with the intranasally vaccinated pigs challenged with the virulent Georgia 2007/1 strain at days 3, 7 and 12 postvaccination. Only the animals of 12 dpv group effectively controlled infection progression, showing low virus loads, minor clinical signs and associated ASFV specific IgG levels. The blood transcriptomic analysis showed lack of the unbalanced inflammatory response, and presence of a cytotoxic response associated with protection from ASFV disease. The study revealed the indicators for immune protection of ASFV during LAV vaccination, provided understanding of ASFV protective immunity and some guidance of ASFV effective vaccine development.

Following questions should be addressed:

1, The authors indicated that the humoral IgG and cytotoxic immunity are the cross-protection indicators for LAV vaccination. Is any difference between LAV induced homologous protection and heterologous protection (cross-protection)? The point should be discussed in the Discussion.

2, In Fig 7B, The blood cytotoxic transcriptomic signatures are enriched at the challenge time in 12 dpv group, indicating protection. Why these signatures decreased at 3 dpc but again increased at 7 dpc in the 12 dpv group? In contrast, in other vaccination groups, the signatures continuously grew during challenge. How to explain it?

3, What is the coating antigen used in ELISA detection of ASFV specific IgG and IgM?

4, There are quite a few inconsistencies between descriptions and the actual results in Figures. These are included as following but not limited, please check thoroughly to correct all the potential errors.

A, the statement of 3/5 survival is not consistent with that in survival curve of Fig 1B which shows 2/5.

B, Where is the moderate in Fig 1 and which color denotes the moderate in Fig 1d? The description is not consistent with that in Materials where five scores from 0 to 4 according to degree of severity (0: normal; 1: mild; 2: moderate; 3 and 4: severe).

C, Based on the data of Fig 2, the BA71ΔCD2 GEC were found in nasal swabs only about day of 10 post-vaccination other than only from 6-7 post-vaccination.

D, Lines 246-247, in the samples obtained postchallenge, Georgia2007/1 related viremia was evident from day 3 dpc in unvaccinated, 3dv and 7dv groups (Figure 2). It is not evident in the 7 dv group.

E, Lines 308-309, in line with the results from the PCA, main transcriptomic changes were observed in group 3dv (757 309 genes) and in the unvaccinated group (1342 genes) at 7 dpc. The main transcriptomic changes should be observed in unvaccinated group and 12 dpv group at 7 dpc, other than between unvaccinated group and 3 dpv at 7 dpc.

Reviewer 3 Report

Comments and Suggestions for Authors

David et al. has studied the protective efficacy of Vaccine Candidate BA71ΔCD2 against virulent Georgia 2007/1 virus strain. African Swine Fever (ASF) is a highly contagious and deadly swine disease that can affect both farm-raised and feral (wild) pigs. The authors have successfully generated results showing a cross protective immunity as early as 12 days post vaccination. Earlier they have demonstrated protection  at 21 days of post vaccination challenge and their study shows induction of virus-specific Th1 and cytotoxic responses is the major CoPs for protection. In this study they have demonstrated protection immunity is conferred by virus-specific IgG. The study results are interesting and there are few suggestions before publication.

1. Elaborate the introduction more about the difference between successful BA71ΔCD2 LAV and other failed LAVs against challenge in terms of immune response.

2. The IgG levels between 7 days and 12 days looks almost similar. Is there any statistical difference between 7 and 12 days vaccination? 

3. Does the authors studied which subclass of IgG is responsible for this protection?

4. If IgG plays major role then passive antibody administration will protect the animals from challenge? Authors need to discuss more about it in discussion with appropriate references.

Round 2

Reviewer 1 Report

Comments and Suggestions for Authors

The authors have responded to all comments and made detailed revisions to the manuscript.